# Effect of Operational Parameters of Unmanned Aerial Vehicle (UAV) on Droplet Deposition in Trellised Pear Orchard

Peng Qi [1,2,3,†], Lanting Zhang [1,2,3,†], Zhichong Wang [4], Hu Han [1,2,3], Joachim Müller [4], Tian Li [1,2,3], Changling Wang [1,2,3], Zhan Huang [1,2,3], Miao He [1,2,3], Yajia Liu [1,2,3,*] and Xiongkui He [1,2,3,*]

1 College of Science, China Agricultural University, Beijing 100193, China
2 Centre for Chemicals Application Technology, China Agricultural University, Beijing 100193, China
3 College of Agricultural Unmanned System, China Agricultural University, Beijing 100193, China
4 Tropics and Subtropics Group, Institute of Agricultural Engineering, University of Hohenheim, Stuttgart 70599, Germany
* Correspondence: liuyajia@cau.edu.cn (Y.L.); xiongkui@cau.edu.cn (X.H.); Tel.: +86-10-6273-2830 (Y.L.); +86-106-273-1446 (X.H.)
† These authors contributed equally to this work.

**Abstract:** Background: Unmanned Aerial Vehicles (UAVs) are increasingly being used commercially for crop protection in East Asia as a new type of equipment for pesticide applications, which is receiving more and more attention worldwide. A new model of pear cultivation called the 'Double Primary Branches Along the Row Flat Type' standard trellised pear orchards (FT orchard) is widely used in China, Japan, Korea, and other Asian countries because it saves manpower and is suitable for mechanization compared to traditional spindle and open-center cultivation. The disease and pest efficacy of the flat-type trellised canopy structure of this cultivation is a great challenge. Therefore, a UAV spraying trial was conducted in an FT orchard, and a four-factor (SV: Spray application volume rate, FS: Flight speed, FH: Flight height, FD: Flight direction) and 3-level orthogonal test were designed. Results: These data were used to analyze the effect, including spray coverage, deposit density, coefficient of variation, and penetration coefficient on the canopy, to determine the optimal operating parameters of the UAV for pest efficacy in FT orchards. The analysis of extremes of variance showed that factor FD had a significant effect on both spray coverage and deposition density. Followed by factor FS, which had a greater effect on spray coverage ($p < 0.05$), and factor SV, FH, which had a greater effect on deposition density ($p < 0.05$). The effects of different factors on spray coverage and deposit density were FD > FS > FH > SV, FD > FH > SV > FS, in that order. The SV3-FS1-FH1-FD3, which flight along the row with an application rate of 90 L/ha, a flight speed of 1.5 m/s, and a flight height of 4.5 m, was the optimal combination, which produced the highest deposit density and spray coverage. It was determined through univariate analysis of all experimental groups, using droplet density of 25/cm$^2$ and spray coverage of 1%, and uniformity of 40% as the measurement index, that T4 and T8 performed the best and could meet the control requirements in different horizontal and vertical directions of the pear canopy. The parameters were as follows: flight along the tree rows, application rate not less than 75 L/ha, flight speed no more than 2 m/s, and flight height not higher than 5 m. Conclusion: This article provides ample data to promote innovation in the use of UAVs for crop protection programs in pergola/vertical trellis system orchards such as FT orchards. At the same time, this project provided a comprehensive analysis of canopy deposition methods and associated recommendations for UAV development and applications.

**Keywords:** unmanned aerial vehicle; droplet deposition; spray coverage; droplet density; orthogonal experiment; pear orchard

## 1. Introduction

Pear is one of the most popular fruits in the world, and it is the third most important fruit in China after apple and citrus. According to the United Nations Food and Agriculture

Organization (FAO), 2021 statistics show that China's pear cultivation area of about 986,479 hectares, the total output of about 18,978,144 tonnes [1], respectively, accounting for 70.5% and of the world's total cultivation area 74.0% of the world's total production, ranking firmly first in the world [2]. China has a long history of pear cultivation, and literature states that pear cultivation in China is more than 3000 years old [3]. The current problem is that the old model for planting fruit trees is slow to update, requiring many labor-intensive, tedious processes that cannot be mechanized [4].

Pest efficacy in orchards is an important industry, and the number of pesticide applications is 8 to 15 times per year [5]. While the frequent chemical application has been effective in controlling pests and diseases, it has led to the '3R' (Residue, Resurgence, and Resistance) problems, which affect the entire agroforestry ecosystem [6]. Over the past decade, many research institutions have designed and developed orchard sprayers as well as pesticide reduction and precision application technologies for orchards [7–11]. For example, the profiling technology changes the spraying parameters in real-time according to the canopy characteristics of the target crop [12,13]. Target application technology that uses sensing and detection technology to spray with trees and not to spray without trees [14–16]. Automatic spraying systems utilize machine vision as well as image processing [17,18]. These can greatly improve the efficiency of pesticide application and reduce the number of pesticides. However, problems such as low operational efficiency and exposure of operators to pesticides posing a health hazard, still exist [19].

In recent years, new pesticide spraying equipment based on unmanned aerial vehicles (UAVs) has been developed in Asia [20,21], especially with the support and leadership of the Chinese government. Meanwhile, there are problems such as labor intensity and labor shortage in orchard cultivation. Many locations limit the use of ground spraying machinery, such as hilly terrain, high-density planting patterns, irregular spacing, and fragmented land [22]. It is also these reasons that have accelerated the development of the UAV industry. Many researchers and enterprises are actively exploring the application of UAVs in orchards.

Scientists have conducted many field trials to study food crops using UAVs, with most research applications related to pesticide spraying by UAVs [23,24], followed by seeding [25,26] and fertilization. And some sensors, RGB cameras, thermal imagers, multispectral, hyperspectral cameras, etc., were carried by UAVs for detection. These were used to collect canopy images [27–30], evaluate the yield [31], assess pruning effects [32], evaluate the economic benefits [33], detect disease [34], and assess spray deposition [35] in modern orchards.

UAVs are less often used in orchards, mainly because of two limitations. (1) Ground orchard sprayers usually spray pesticides at an application rate of 800–1500 L/ha, while UAVs carry less than 30 L. If UAVs operate at the same pesticide rate as ground sprayers, they need to take off and land frequently, which seriously affects the operational efficiency of UAVs. (2) The downwash airflow of the UAV moves vertically down from the top of the tree canopy, preventing contact of the spray droplets with the adaxial side of the leaves (opposite of the ground sprayer), so spray coverage, and deposit density do not meet pest efficacy requirements. At the same time, the downwash airflow is blocked by the larger tree canopy, resulting in an unstable flow distribution. Therefore, if the vertical component of downwash airflow is weak, the penetration will be weakened, and the risk of drift loss will be increased [36].

A large number of pesticides are widely used to control various pests and diseases in order to improve the yield and quality of agricultural products [37]. The complex cultivation structure of orchards leads to different pesticide requirements. In order to achieve accurate pesticide application, it needs to be simulated according to the canopy structure [38]. For ground orchard sprayers, a number of studies have shown that target-specific profiling variable sprays are used to adjust the profiling mechanism of the profiling sprayer to match the contours of the tree canopy [39–42]. However, due to space and load constraints, UAVs cannot carry heavy equipment [43]. Therefore, in order to achieve precise

pesticide application and up to effectiveness pest efficacy, more solutions are needed to adjust the operating parameters of the UAVs [44].

Numerous studies have been conducted with the aim of quantifying the relationship between the quality of the spray application process and the differences in canopy characteristics [45,46]. Different shapes of the canopy structure and different operating parameters have important effects on the deposition of droplets [47]. Derksen et al. [48] suggest experiments with different application rates and speed settings that can make effective applications more efficient. Trees with an open tree center can achieve a higher density of droplet deposition than those with a rounded crown shape. The UAV performed better on open center-shaped plants at a working height of 1.0 m [49]. The deposit density in the lower layer of inverted triangle-shaped trees was 48.04% higher than in triangle-shaped trees [50]. The spraying uniformity is different between the Y-shape and central-leader-shape peach trees. In trees with a Y-shape, droplets are distributed more uniformly in both the inner and outer layers [51]. UAV spraying at a flight height below 1.0 m and a flying speed of 1.7 m/s with an open tree shape were able to achieve better droplet penetration and distribution in citrus orchards [52]. The effect of the inverted triangular shape on the lower droplet density was more pronounced, showing an 82.0% increase in droplet density compared to the triangular shape [53].

With the development of the pear industry, the problem of soaring labor costs is becoming more and more prominent, and saving manpower has become a research and production imperative [54]. In 2008, the 'The modern agricultural industry technology system' construction special was launched by the Chinese government [55]. The technology of pear cultivation was carried out in more comprehensive, systematic, and thorough research. The weak point of orchard mechanization is constantly broken by the integration of agricultural machinery in the field. Pear cultivation is constantly innovated, and the cultivation of pear trees is changing or optimized [56].

A unique cultivation pattern has been invented by the Hubei Province Academy of Agricultural Sciences Fruit and Tea Research, which is known as 'Double Primary Branches Along the Row Flat Type' standard trellised pear orchards **(FT orchard)** [57]. This structure is easier to manage and mechanize than the traditional treeless cultivation model and solves the 'trellis separation' problem of existing three-pole trellises [58,59]. Moreover, the **FT orchard** is conducive to enhancing photosynthesis and the accumulation of organic matter, which helps to improve the quality and yield of fruits [60].

There is no suitable application technology for the new cultivation methods, and the traditional application method is prone to many problems, such as excessive application, heavy pollution, and pesticide residues [61]. In this study, based on the previous work, a 3-level orthogonal test with four factors (spray application volume rate, flight speed, flight height, and flight direction) was designed using a representative model of a multi-rotor UAV. The parameters of the multi-rotor UAV for pear orchard trellis operation were preferentially selected by correlation analysis of the orthogonal test results. It is expected to provide supplementary information to improve UAV field operation parameters for trellised pear orchards and droplet deposition criteria, provide a reference for the preferred selection of field operation parameters of other similar models, and provide a basis for developing technical specifications of orchard operations based on agricultural unmanned aerial vehicles.

## 2. Materials and Methods

### 2.1. Field Plots

Two field experiments were conducted in October 2021 at the Shanxi Province Agricultural Academy Fruit Tree Institute (112°35′24″ E, 37°25′51″ N) and in September 2021 at the Hubei Province Agricultural Academy Fruit Tree Institute (114°19′27″ E, 30°29′14″ N), China (Figure 1). Both experimental fields were planted with **FT orchards**. This was a very special cultivation method in which the canopy was all concentrated on the top of the trunk, and the branches were woven into a grid using special agricultural methods, as

shown in Figure 2. Pear trees were cultivated in a row spacing of 3 m and a tree spacing of 4 m. The trees in the basic FT orchard formed conjoined rows with a height of 3.0–3.5 m and a crown diameter of 0.5–0.8 m.

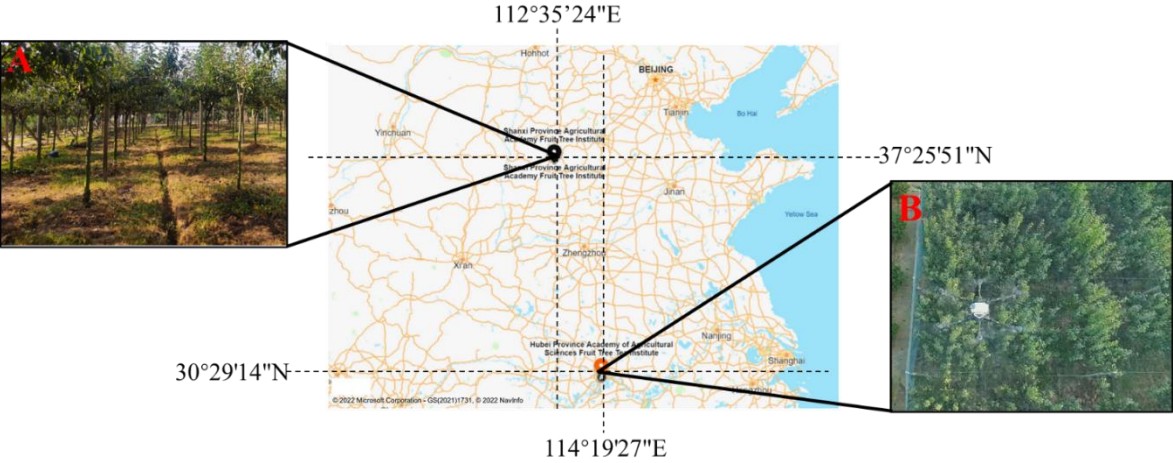

**Figure 1.** Experimental location in Hubei and Shanxi Province. (**A**) represents the test site at Shanxi Province Agricultural Academy Fruit Tree Institute, and (**B**) represents the test site at Hubei Province Agricultural Academy Fruit Tree Institute.

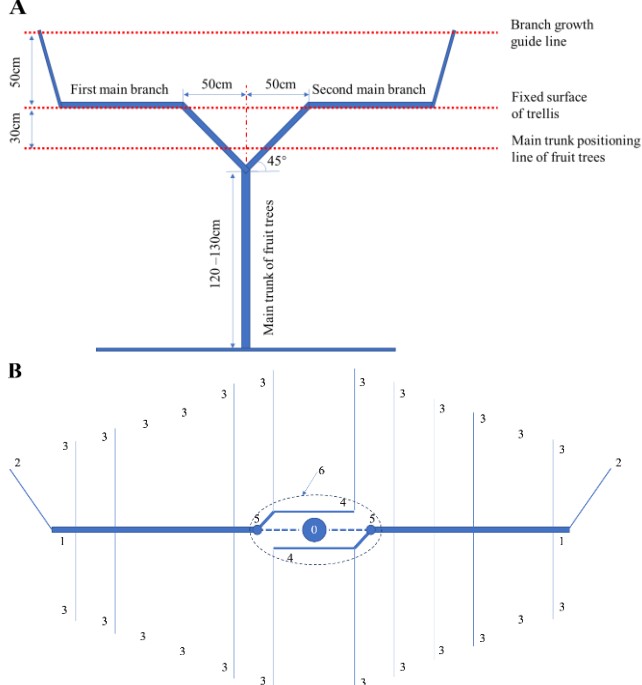

**Figure 2.** FT orchard tree shape results from the distribution of branch groups. (**A**) represents the front view and the distribution of the main branch structures. (**B**) represents the top view, which mainly shows various branch growth structures, where 0 represents the main stem, 1 represents the main branch, 2 represents the main branch extension, 3 represents the group of branches growing fruit, 4 represents the group of F branches growing fruits, 5 represents the fixed connection point with the bracket, 6 represents funnel-shaped space.

### 2.2. Spraying Platform and Spraying Systems

The unmanned aerial vehicle was a series UAV sprayer (DJI T20, SZ DJI Technology Co., Ltd., Shenzhen, China), as shown in Figure 3.

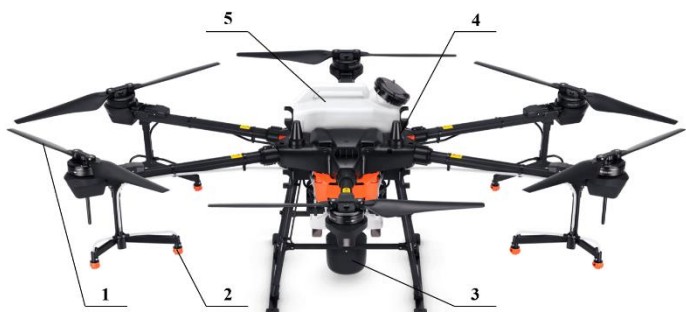

**Figure 3.** DJI T20 series plant protection UAV. (1) represents the propeller, which provides lift, (2) represents the nozzle, the droplet atomization device, (3) represents omnidirectional radar, which senses obstacles, (4) represents D-RTK, which is used for localization, (5) represents tank, which is used to hold the pesticide solution.

The UAV was equipped with a GNSS + RTK dual redundancy system (SZ DJI Technology Co., Ltd., Shenzhen, China) that provided centimeter-level high-precision positioning, while omnidirectional digital radar was installed to provide horizontal omnidirectional obstacle sensing and horizontal omnidirectional obstacle bypassing functions, which could plan obstacle avoidance paths, automatically bypass obstacles, and support ground-like flight.

A variable pesticide application system was installed on the UAV. The spray flow rate was automatically adjusted according to the flight speed during the operation of the UAV. A four-channel electromagnetic flow meter (SZ DJI Technology Co., Ltd., Shenzhen, China) ensured that the flow rate was consistent from nozzle to nozzle. All adjustments were made via software that controlled application rate, flight speed, and relative flight height. Two types of nozzles (SX11001VS/SX110015VS, Spraying Systems Co., Wheaton, IL, USA) were selected for aerial crop protection in this experiment. Eight extended-range flat-fan nozzles (Spraying Systems Co., Wheaton, IL, USA), divided into four sets, were attached below the corresponding rotors of the UAV, which rendered better atomization stability. The spraying system also comprised auxiliary components such as a peristaltic pump and tank. The technical parameters of the UAV are presented in Table 1.

**Table 1.** T20 UAV technical parameters.

| Classification | Parameters |
| --- | --- |
| Size (m) | $2509 \times 2213 \times 732$ |
| Tank volume (L) | 20 |
| Nozzle number | 8 |
| Flow range (L/min) | 0–6 |
| Droplet size (μm) | 130–300 |
| Spray width (m) | 4–7 |
| Flight precision, D-RTK (m) | Horizontal $\pm$ 0.10, vertical $\pm$ 0.10 |
| Flight speed (m/s) | 0–7 |

In the previous research, the precise measurement of the droplet size spectrum of the flat-fan nozzle was obtained following the method previously described by [62–66], which was carried out using a laser diffraction system (SprayTec, Malvern Panalytical Ltd., Malvern, Worcestershire WR14 1GD, UK) at the Centre for Chemicals Application Technology, China Agricultural University according to ISO standard [67].

*2.3. Experimental Design*

To study the effect of application parameters on spray deposition distribution, based on the orthogonal experimental design method, a 4-factor 3-level orthogonal test was designed to investigate the effects of application rate, flight height, flight speed, and flight direction

on droplet deposition in trellised pear canopies to determine appropriate parameters for UAV application in **FT orchard**. These factors and levels are described in Table 2.

**Table 2.** Factors and levels settings of the orthogonal test.

| Level | Factor | | | |
|---|---|---|---|---|
| | **(SV) Spray Volume (L/ha)** | **(FS) Flight Speed [m/s]** | **(FH) Flight Height [m]** | **(FD) Flight Direction \*** |
| 1 | 60 | 1.5 | 4.5 | P |
| 2 | 75 | 2.0 | 5.0 | V |
| 3 | 90 | 2.5 | 5.5 | P |

\* P stands for: route parallel to pear tree row, V stands for: route across to pear tree row.

All parameters were in juxtaposition, and interactions between factors were not examined. The orthogonal table $L_9(3^4)$ was used to arrange the tests according to the factors and levels examined in the tests. The detailed tests are presented in Table 3. Considering the complexity of the actual test environment, the number of tests was minimized, but effective repetition should be guaranteed. If strong gusts or severe course deviations were observed during the test, these should be regarded as invalid data. The spray solution was pure water, and three valid replicates were performed in each treatment group.

**Table 3.** Orthogonal test design.

| Treatment | Factor SV | Factor FS | Factor FH | Factor FD |
|---|---|---|---|---|
| T1 | 1 (60) | 1 (1.5) | 1 (4.5) | 1 (P) |
| T2 | 1 (60) | 2 (2.0) | 2 (5.0) | 2 (V) |
| T3 | 1 (60) | 3 (2.5) | 3 (5.5) | 3 (P) |
| T4 | 2 (75) | 1 (1.5) | 2 (5.0) | 3 (P) |
| T5 | 2 (75) | 2 (2.0) | 3 (5.5) | 1 (P) |
| T6 | 2 (75) | 3 (2.5) | 1 (4.5) | 2 (V) |
| T7 | 3 (90) | 1 (1.5) | 3 (5.5) | 2 (V) |
| T8 | 3 (90) | 2 (2.0) | 1 (4.5) | 3 (P) |
| T9 | 3 (90) | 3 (2.5) | 2 (5.0) | 1 (P) |

### 2.4. Sampling Layout

According to the structural characteristics of the **FT orchard** cultivation, the canopy was divided into an upper and lower part in the vertical direction. The upper part represented the newly grown branches of the pear tree, also known as the "nutrient branch", which are relatively short and not woven into a grid, hereinafter collectively called the "nutrient layer" **(NL).** The lower part represents the mature branches, which were already woven into a grid. It represents the area of fruit growth, hereinafter collectively called the "fruit layer" **(FL)**, as shown in Figure 4A.

According to the different zones of the canopy and ISO22522 [68], each target tree was divided into three levels: NL deposition, FL deposition, and ground loss. As shown in Figure 4B, the *X*-axis ran in parallel to the tree row, the *Y*-axis was across to the tree row, and the *Z*-axis was vertical to the ground. In the upper and lower part of the NL, in each layer of the fruit tree, it was divided into five sampling points, which were front (*X*-axis positive direction), back (*X*-axis negative direction), left (*Y*-axis positive direction), right (*Y*-axis negative direction) and middle (*Z*-axis direction), the sampling points were placed symmetrically at 2 m intervals along the *X*-axis and at 1.5 m intervals along the *Y*-axis. The sampling points were placed the same way in the FL. Two pieces of water-sensitive paper (WSP) (38 × 26 mm, Syngenta Crop Protection AG, Basel, Switzerland) were attached at each sampling point using a clip. They were fixed along the petiole to the adaxial and abaxial surfaces of the leaf so that the collectors fitted closely to the surface of the leaf to ensure that the droplets received by the collectors were at a similar angle to the leaf,

to evaluate spray coverage parameters on the surface of the leaves. Front-up WSP cards were placed on the ground at five corresponding positions below the canopy to assess the ground loss.

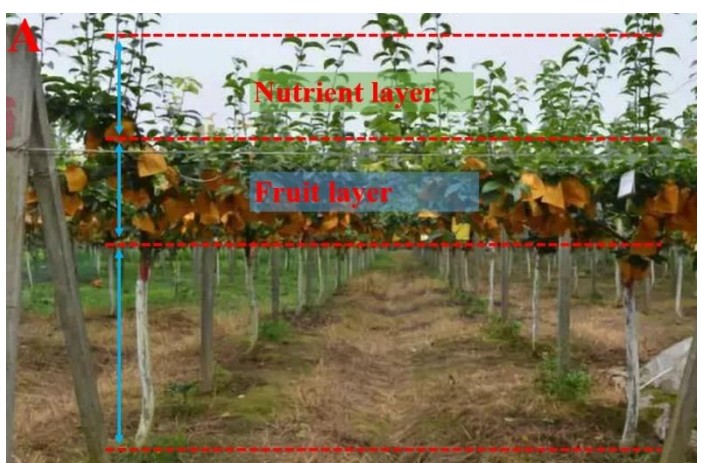
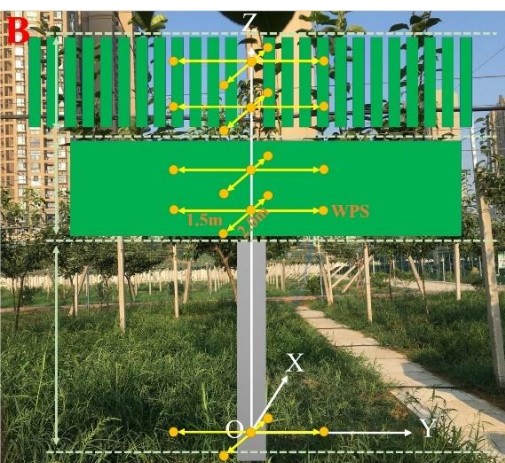
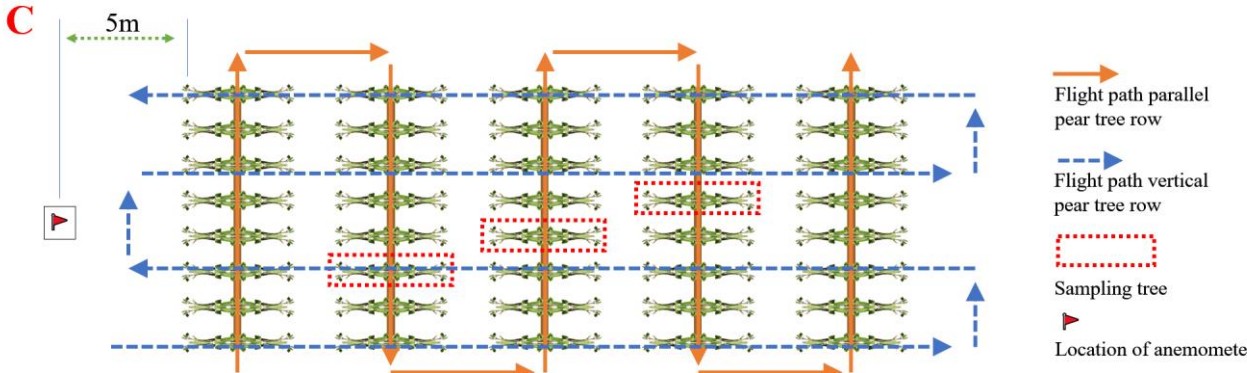

**Figure 4.** Sampling locations for assessing spray deposition. (**A**) represents the structural characteristics of the **FT orchard** cultivation, (**B**) represents sampling locations in the target pear tree canopy and ground, and (**C**) represents **a** schematic diagram of site layout and flight patterns in **FT orchard** tests.

In the **FT orchard**, the test area was chosen away from the boundary to ensure that the UAV downwash airflow could be steadily maintained in the test area. Three pear trees were selected as a target for sampling points to capture spray deposition in the test area Figure 4C.

*2.5. Weather Conditions*

The environmental metrological conditions were tested using an anemometer (Pocketwind IV, Lechler GmbH, Metzingen, Germany) with reference to the ISO 22,522 standard. Details are shown in Table 4. The anemometer was about 5 m upwind of the test area (Figure 4C).

**Table 4.** Meteorological conditions during the test.

| Test Location | Temperature [°C] | Humidity [%] | Wind Speed [m/s] | Wind Direction [°] |
|---|---|---|---|---|
| Hubei | 28–31 | 55% | 0.8–1.5 | 234 |
| Shanxi | 13–16 | 64% | 0.2–1.2 | 200 |

*2.6. Characterization of the Spray Deposition and Statistical Analysis*

Each test was completed after the droplets on the WPS had dried, and the WPS were attached to labeled papers and stored in a sealed bag to avoid moisture contamination. According to the method of [69], all collected WSP samples were scanned using a scanner (EPSON DS-1610, Seiko Epson Corporation, Nagano-ken, Japan) at high pixel (600 dpi × 600 dpi) resolution. Then, the scanned pictures were determined using the macro DepositScan programmed in ImageJ software (V1.38x, National Institutes of Health, Bethesda, Maryland, USA) to calculate the data of droplet deposition parameters, such as the spray coverage, droplet deposit density, and droplet size.

Spray deposition parameters were obtained for canopy location, as follows:

(1) Spray coverage (SC, %). The ratio of the target surface area covered by droplets to the total target surface area.

(2) Deposit density (DD, deposits/cm$^2$). The number of spray deposits per unit surface area (usually 1 cm$^2$).

(3) DV50 (μm) is the particle size below which 50% of the spray lies. This may be termed the "fifty percent cut-off point".

In addition, there were two parameter values that indicated droplet distribution performance.

(4) The coefficient of variation (CV, %) indicated the uniformity of the deposition distribution of the spray coverage parameters on the canopy:

$$CV(\%) = \frac{SD}{\overline{X}} \times 100 \tag{1}$$

where *SD* is the standard deviation of the sample and $\overline{X}$ is the average coverage parameters of the droplets, with:

$$SD = \sqrt{\frac{\sum_{i=1}^{n} (X_i - \overline{X})^2}{(n-1)}} \tag{2}$$

where $X_i$ is the droplet coverage of each sampling point and n is the number of sampling points of each test group.

A lower CV means that spray coverage is distributed more evenly. According to the sampling points of pear trees, this work distinguished between the uniformity of deposition distribution in the horizontal direction and the uniformity of deposition distribution in the vertical direction, which were denoted by (CV_P, %) and (CV_V, %), respectively.

(5) The penetration coefficient (PC, %) is the percentage of droplets collected on the fruit layer in the vertical direction of the pear tree to the total number of droplets in this direction. A higher PC value indicates better spray penetration.

In this research, statistical analyses were performed using SPSS Statistics (Version 26, IBM Corporation, Armonk, NY, USA), one-way ANOVA analysis, and multi-factor main effects analysis to establish the effect of treatment with LSD and Duncan's post hoc test at a significance level of 0.05.

*2.7. Comprehensive Evaluation Methods and Evaluation Criteria*

Agrochemicals in fruit orchards are usually applied with an air-assisted sprayer, which is a conventional volume sprayer with an application rate of at least 450 L/ha. Droplet disposition quality is usually assessed by two elements, namely SC and DD. According to the NY/T992 [70] standard requirements, SC shall not be less than 33%, DD is greater than or equal to 30 deposits/cm$^2$ for systemic pesticides, and DD is not less than 70 deposits/cm$^2$ for general pesticides.

However, spray application rates of 7.5–450 L/ha are used as a common range for sprayers [71,72]. In particular, less than 50 L/ha is used as the application rate for UAVs, and sometimes an application rate of 15 L/ha achieves good results in orchard applications [73]. For the standard of ground sprayers, the application rate of UAV is a low-volume spray.

In the existing standards for UAV applications, there is only one indicator for DD and no indicator for SC.

In the standards [71,74], there are different DD requirements for different pesticide formulations (shown in Table 5), but the coverage density allowed is not less than 20 deposits/cm$^2$.

**Table 5.** Droplet density quality standard from NY/T650.

| Title | Pesticide Variety | | Index | |
|---|---|---|---|---|
| | | | Normal-Volume $\geq$450 L/ha | Low-Volume 7.5–450 L/ha |
| SC, % | Non-systemic pesticide | | $\geq$33 | / |
| DD,/cm$^2$ | Insecticide | | / | $\geq$25 |
| | Fungicide | Systemic | / | $\geq$20 |
| | | Non-systemic | / | $\geq$50 |
| | Herbicide | Systemic | / | $\geq$30 |
| | | Non-systemic | / | $\geq$50 |

Until 2018, the standard published on the technical specification of quality evaluation for crop protection UAV was based on using only 15 deposits/cm$^2$ as the minimum droplet density to measure the effective spray width [75].

In orchard studies, some researchers used SC 33% as an evaluation indicator [76], and others used 15 deposits/cm$^2$ as an evaluation indicator [77,78].

Although droplet size has an effect on droplet deposition [79], the fact that droplets repeatedly fall on the same spot makes measurement data very difficult, and there is no standard to follow. This has resulted in fewer researchers using deposit density as an evaluation metric. In the quality indexes of the agricultural aviation operation standard [80], the droplet size is required to be within a certain range, usually 150–300 μm for insecticides and fungicides at an application rate of 5 L/ha or more.

DD was used as the most important evaluation index when measuring the spray quality of UAVs. As M. Salyani and R. D. Fox [81] proved that percent area appeared to be the most reliable, SC was used as a secondary evaluation index. DV50 was not used as an evaluation indicator but only as an observation indicator. According to the study in the previous section, the DD was at least 15 deposits/cm$^2$. Wang et al. [82] used 15 deposits/cm$^2$ as the determination index and 1% SC as the detection index. The groups were evaluated according to this criterion in this study.

Currently, UAVs are mainly used for spraying systemic insecticides and fungicides. Combined with the previous analysis, a coverage rate of 1% and droplet density of 25 deposits/cm$^2$ are used as evaluation criteria in this paper.

According to the requirements for DD, SC, and CV, we define the standard and perfect values of completion, as shown in Table 6.

**Table 6.** Weight coefficient of standard and perfect values.

| Level | Values | | |
|---|---|---|---|
| | DD (%) | SC (Deposits/cm$^2$) | CV (%) |
| Standard | 25 | 1 | 40 |
| Perfect | 50 | 10 | 0 |

"Standard" represents the value specified in the standard, while "Perfect" represents exceeding or achieving 100% of that value.

$$DDw_i = (DD_i - 25)/(50 - 25) \tag{3}$$

$$SCw_i = (SC_i - 1)/(10 - 1) \tag{4}$$

$$CVw_i = (CV_i - 0)/(40 - 0) \tag{5}$$

$$Com\_E_i = DDw_i + CV\_V\_DDw_i + CV\_P\_DDw_i + SCw_i + CV\_V\_SCw_i + CV\_P\_SCw_i \tag{6}$$

$DDw_i$, $SCw_i$, $CVw_i$, where "*w*" represents the weights and "*i*" represents the ordinal number of trials 1–9. $\_V\_DDw_i$, $\_P\_DDw_i$, where "*P*" stands for horizontal direction and "*V*" stands for the vertical direction.

In this paper, the combined DD and SC were used as a comprehensive score to evaluate the quality of droplet deposition (Com_E). In Equation (6), *DD*, *SC*, and *CV* are three independent indices to evaluate the quality of droplet deposition, and their three values are normalized by Equations (3)–(5), respectively, so that the indices are in the same order of magnitude. *DD* and *SC*, as indexes for intuitively evaluating the quality of droplet deposition, the uniformity of distribution of *DD* and *SC* in the canopy also affects the effect of final biological control, so the *CV* is included in the comprehensive evaluation as a similar correction term.

## 3. Results and Discussion

### 3.1. Effect of Different Factors on the Deposition of Droplet in the Canopy

3.1.1. Range Analysis of Different Factors

All samples were collected as overall canopy data for extreme difference analysis, which was performed by analyzing the UAV flight speed, height, direction, and application rate that mainly affect the droplet deposition. Table 7 indicates the results of the range analysis among the different factors. The primary and secondary order factors that affected SC, DD, and DV50 were found to be FD > FS > FH > SV, FD > FH > CV > FS, and FH > FS > SV > FD in that order. From the range analysis, the best combination to improve SC was SV3-FS1-FH1-FD3, the best combination to improve DD was SV3-FS1-FH1-FD3, the best combination to improve DV50 was SV2-FS1-FH3-FD2.

**Table 7.** Results of Range Analysis.

| | SC, % | | | | DD, Deposits/cm² | | | | DV50, μm | | | |
|---|---|---|---|---|---|---|---|---|---|---|---|---|
| | Factor SV | Factor FS | Factor FH | Factor FD | Factor SV | Factor FS | Factor FH | Factor FD | Factor SV | Factor FS | Factor FH | Factor FD |
| K1 | 385.14 | 485.41 | 506.86 | 284.28 | 12,111.90 | 13,869.10 | 17,842.00 | 8012.94 | 48,877.00 | 62,757.50 | 45,760.00 | 51,857.50 |
| K2 | 450.24 | 481.06 | 424.59 | 303.91 | 11,241.10 | 13,686.60 | 11,913.54 | 12,110.00 | 51,486.00 | 55,704.00 | 46,077.00 | 62,065.00 |
| K3 | 486.23 | 355.14 | 390.16 | 733.42 | 17,069.14 | 12,866.44 | 10,666.60 | 20,299.20 | 64,678.00 | 46,579.50 | 73,204.00 | 51,118.50 |
| k1 | 1.07 | 1.35 | 1.41 | 0.79 | 33.64 | 38.53 | 49.56 | 22.26 | 135.77 | 174.33 | 127.11 | 144.05 |
| k2 | 1.25 | 1.34 | 1.18 | 0.84 | 31.23 | 38.02 | 33.09 | 33.64 | 143.02 | 154.73 | 127.99 | 172.40 |
| k3 | 1.35 | 0.99 | 1.08 | 2.04 | 47.41 | 35.74 | 29.63 | 56.39 | 179.66 | 129.39 | 203.34 | 142.00 |
| Range | 0.28 | 0.36 | 0.32 | 1.25 | 16.19 | 2.79 | 19.93 | 34.13 | 43.89 | 44.94 | 76.23 | 30.41 |
| Optimum Level | SV3 | FS1 | FH1 | FD3 | SV3 | FS1 | FH1 | FD3 | SV2 | FS1 | FH3 | FD2 |
| Order | | FD > FS > FH > SV | | | | FD > FH > SV > FS | | | | FH > FS > SV > FD | | |

In this table, K is the sum of factor test results; k is the mean value of the sum of factor test results; and the Range is the larger of the k values minus the smaller. The optimum level of each factor can be determined according to the size of the k value. The influence order of different factors is determined by the range value.

3.1.2. Variance Analysis of Different Factors

The orthogonal test range analysis is relatively intuitive and simple. Still, it cannot distinguish whether the differences in the indicators are caused by changes in the test factors or by errors in the test [83]. Various factors can lead to differences in the data between groups, and errors can also lead to differences in the data of the same group, so ANOVA was used to test the significance of the influences.

The droplets collected from each tree were subjected to ANOVA of the primary factors of the four factors, as shown in Table 8. The SC was mainly significantly different in factor FS and factor FD. The primary order of the factors' influence on SC was FD > FS > FH > SV. The DD was significantly different, mainly in factor SV, factor FS and factor FD. The primary order of the factors' influence on DD was FD > FH > SV > FS. The DV50 was significantly different between all factors. The primary order of the factors' influence on DD was FH > FS > SV > FD.

**Table 8.** Results of Primary Factors Analysis.

| Dependent Variable | Sig. | | |
|---|---|---|---|
| | SC, % | DD, Deposits/cm$^2$ | DV50, μm |
| Factor SV | 0.287 | 0.000 | 0.000 |
| Factor FS | 0.002 | 0.562 | 0.000 |
| Factor FH | 0.108 | 0.000 | 0.000 |
| Factor FD | 0.000 | 0.000 | 0.000 |

Statistical significance level: Duncan test $p < 0.05$, 0.000 represents $p$ value lower than $10^{-4}$.

All factors were analyzed using the method of range and analysis of variance of the orthogonal test to identify the main factors. Factors FD and FS had a significant effect on SC, and the effect of factor FD was greater than that of factor FS. The trend of DD was different from the trend of SC, where the main difference was factor FD. The factors FD, SV, and FH had a significant effect on DD, with significant effects in the order of factor FD > FH > SV. All factors had a significant effect on DV50, and each factor had a different effect on DV50 than SC and DD.

*3.2. Overall Deposition Characteristics of the Different Test Groups*

3.2.1. Deposition Characteristics of the Total Canopy

Figure 5 shows the DD results of different test groups. DD is mainly divided into four classes ($p < 0.05$), and T8 had the largest value of 81.12 deposits/cm$^2$. T1, T3, T4, T6, and T7 had the second-highest DD with an average of 47.34–31.9 deposits/cm$^2$. The third-ranked was T2 and T9, with an average of 27 deposits/cm$^2$. T5 was the worst, with a DD of only 9.6 deposits/cm$^2$. Except for T5, all other groups had a DD of 27 deposits/cm$^2$ or more.

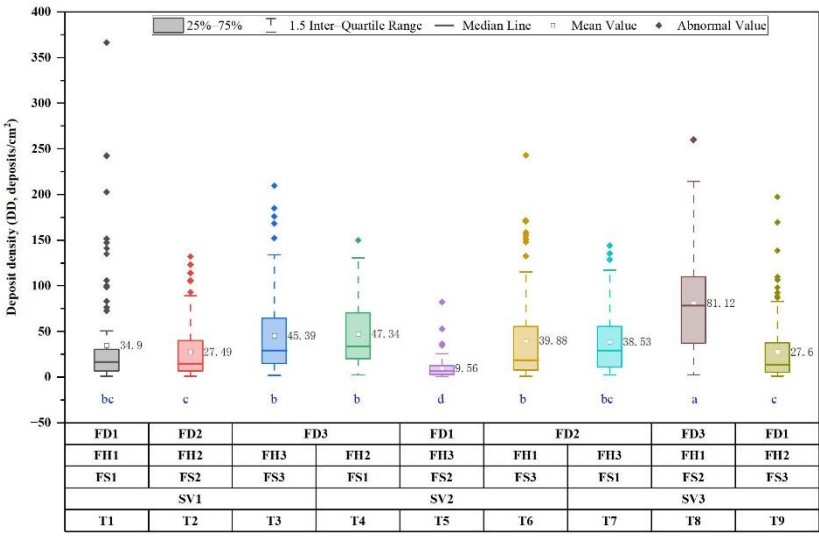

**Figure 5.** Deposition characteristics of droplet density (DD) in the canopy. Numbers in the chart are mean values, and different lowercase letters indicate significant differences at $p \leq 0.05$.

Using SC as an evaluation indicator (Figure 6), T8 and T4 showed the best performance ($p < 0.05$), which was significantly higher than other test groups, reaching 2.67% and 2.14%, respectively. There was no difference between groups T1, T3, and T7, with a coverage of 1.01–1.53%. The other four test groups, namely T2, T5, T6 and T7, where SC was under 1%, fluctuated between 0.66% and 0.83%.

Droplet sizes in all test groups were in the range of 121.0–304.7 μm (Figure 7). And the results of the laboratory tests showed that the droplet size remained between 130–200 μm in the pressure range of 2–4 bar, containing two classes, one fine (F) and the other very fine (VF) [84,85]. Droplet particle size exceeded 200 μm for T5 and T7. This may be due to the

repeated spraying of droplets onto an overlapping point [86]. Another important reason was that the droplet size displayed by WSP was impacted by the droplet, and the droplet size was larger than that obtained in the laboratory.

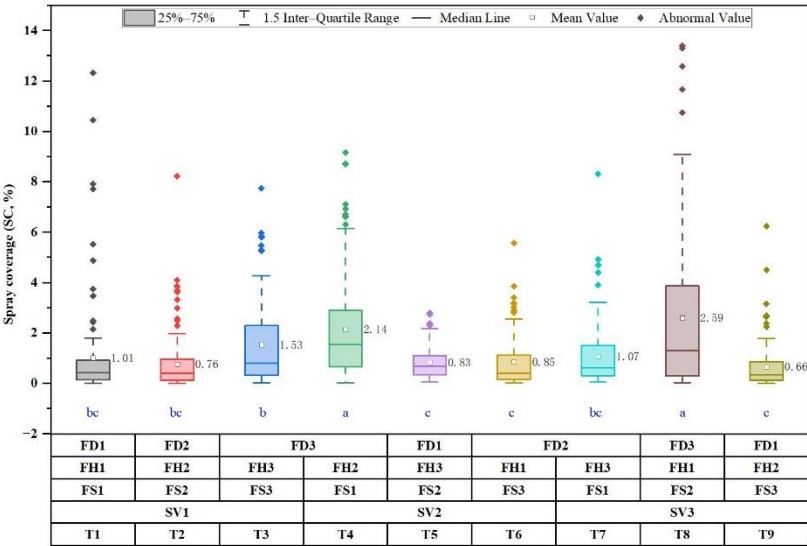

**Figure 6.** Deposition characteristics of spray coverage (SC) in the canopy. Numbers in the chart are mean values, and different lowercase letters indicate significant differences at $p \leq 0.05$.

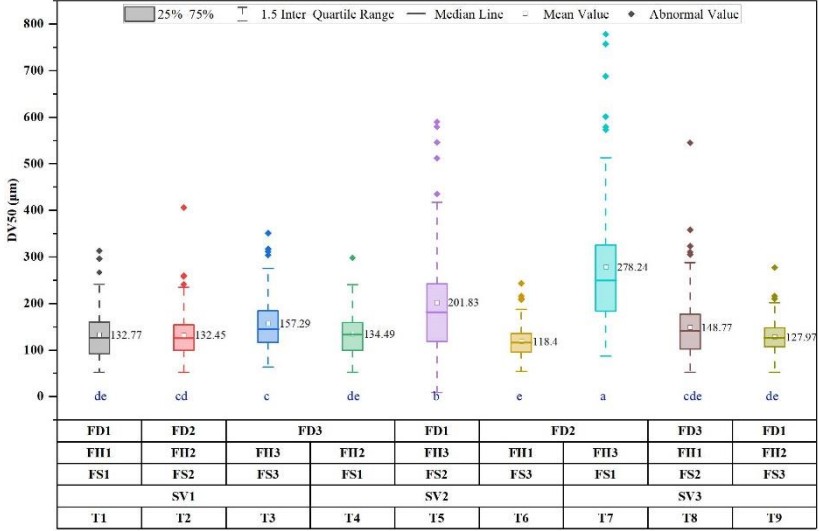

**Figure 7.** Deposition characteristics of DV50 in the canopy. Numbers in the chart are mean values, and different lowercase letters indicate significant differences at $p \leq 0.05$.

### 3.2.2. Droplet Deposition Characteristics on Both Sides of the Leaves

For DD, the overall trend was the same on both sides of the leaves as in the canopy (Figure 8). Among them, DD of T8, T4, and T3 showed the best performance with 117.93 deposits/cm$^2$, 68.6 deposits/cm$^2$, and 60.73 deposits/cm$^2$ on the adaxial side and 43.69, 25.73, and 28.92 on the abaxial side. However, the DD on the abaxial side of the leaves in the other group was less than 25 deposits/cm$^2$, which did not meet the minimum pest and disease prevention and control requirements.

The SC of the adaxial leaves was consistent with the overall trend of the canopy (Figure 9), with T8, T4, and T3 still performing the best, with 4.62%, 3.45%, and 2.27%, respectively. The SC of the abaxial side of the leaves differed from the overall trend. The SC of T4, T3, and T8 was the highest ($p > 0.05$), which was up to 0.65–0.84%. However,

it is extremely critical that all tested groups did not reach 1% SC on the abaxial side of the leaves.

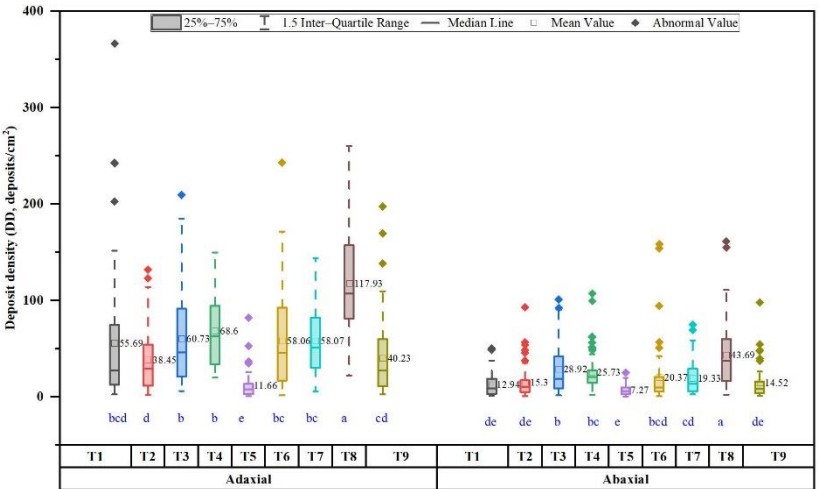

**Figure 8.** DD characteristics on both sides of the leaves. T1 to T9 represent different test groups. Numbers in the chart are mean values, and different lowercase letters indicate significant differences at $p \leq 0.05$.

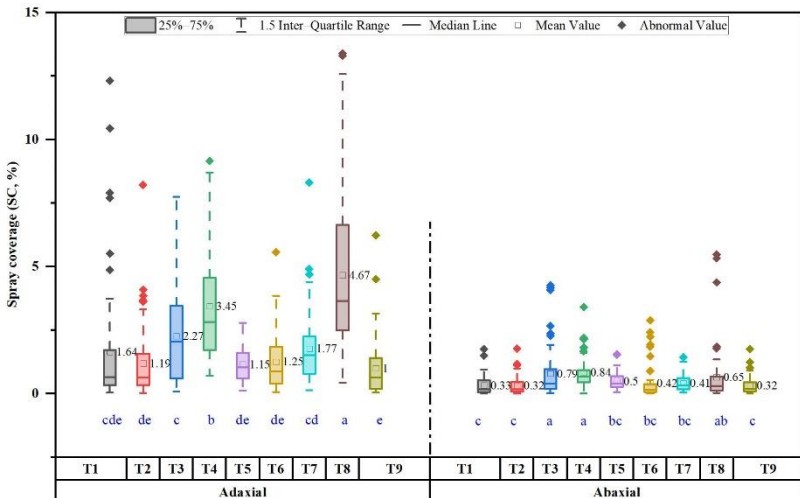

**Figure 9.** SC characteristics on both sides of the leaves. T1 to T9 represent different test groups. Numbers in the chart are mean values, and different lowercase letters indicate significant differences at $p \leq 0.05$.

There was a significant difference in the effect of DD and SC on the adaxial and abaxial sides of the leaves. Although deposition on the abaxial side was usually low, there was an important biological effect of increasing the deposition on the abaxial of the leaf [87]. All test groups did not significantly improve the deposition on the abaxial side of the leaves. All groups failed to meet the standard when 1% of SC was used as the evaluation criterion, and T1, T2, T5, T6, T7, and T9 failed to meet the standard when 25 deposits/cm$^2$ of DD were used as the evaluation criterion. Therefore, a comprehensive evaluation method is needed for UAV operation, which cannot be evaluated by a single index.

It was also found that during the top-down motion of the UAV wind field, droplets were mainly deposited on the adaxial side of the leaf, while the deposition on the abaxial side of the leaf was not improved. The wind field of the UAV was directed vertically downward [88], with droplet deposition concentrating on the upper part of the canopy [89], mainly on the adaxial side of the leaves. This was the opposite trend of the radial spray

pattern of the air-assisted ground sprayer. Therefore, subsequent research should focus on how to improve deposition on the opposite side of the leaves [90].

### 3.2.3. Horizontal Droplet Deposition Characteristics of the Canopy

There were five positions in the horizontal direction of the canopy, namely front, back, left, right, and center of the pear orchard canopy.

The DD of all groups in all directions met the standard requirements except for T1, T2, T5, and T9, where the DD was lower than 25 deposits/cm$^2$ (Figure 10). In T5, the highest DD was found on the left side ($p < 0.05$). In all tests, T4, T5, and T9 were different in all five directions, and the right side was the lowest ($p < 0.05$).

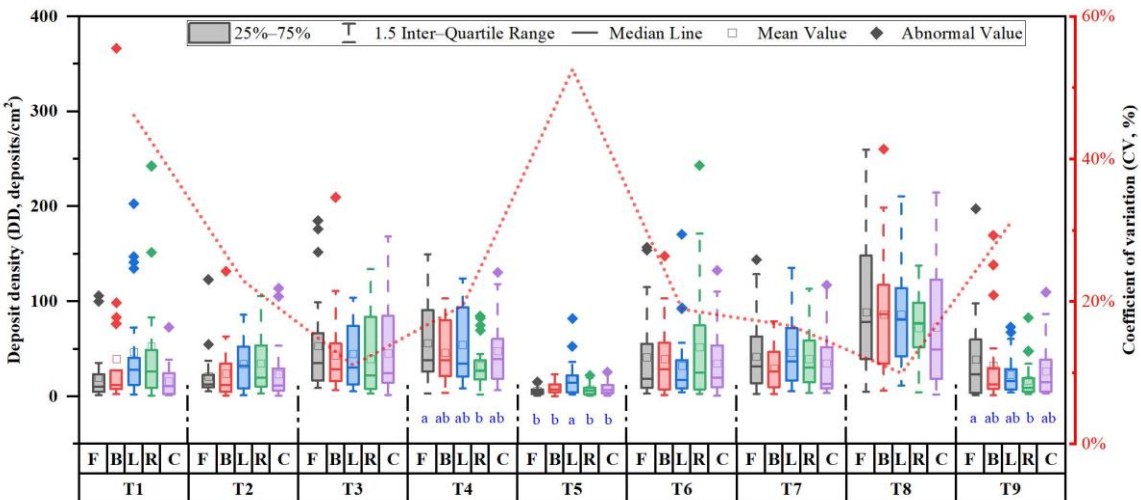

**Figure 10.** Deposition characteristics of DD in the horizontal of the canopy. F represents the front of the pear tree canopy, B represents the back, L represents the left, R represents the right, and C represents the middle of the canopy. Different lowercase letters indicate significant differences at $p \leq 0.05$. The box plot represents DD, and the dashed line represents CV.

For each direction of the canopy, only T3, T4 and T8 met an SC value greater than 1% (Figure 11). And only T8 differed in different directions, again reflecting that the SC was higher on the left than on the right side ($p < 0.05$).

The dashed lines in Figures 10 and 11 represent the CV_P of the DD and SC in the five horizontal directions, respectively.

Except for T1 and T5, the CV_P of DD was below 40%, which met the criteria for UAV operations [75]. T3, T4, and T8 had CV_P of DD in the order of 11%, 19%, and 10%.

The CV_P of SC was less than 40%, except for T1. Meanwhile, T3, T4, T8 had CV_P of SC of 17%, 18%, 39% in that order. However, the CV_P of T8 was as high as 39%, which was similar to the specified standard value.

As shown in Table 9, the effect of the factor FD > SV > FS > FH on CV_P decreased sequentially. FD3-SV3-FS3-FH2 was the best operational parameter to obtain the minimum CV_P. The CV_P of SC was affected in the order FH > SV > FS > FD, and the best parameter of operation was FH3-SV2-FS3-FD3.

The main factor affecting the distribution of droplets is the downwash airflow under the UAV, which consists of the air turbulence generated by the rotor and the wind field of the external environment [91]. The maximum variation coefficient of spray deposition is strongly influenced by wind speed, nozzle position, release flight height, flight speed, and droplet size [92]. When multiple rotor wind fields from the UAV and the ambient wind field overlapped, the wind field did not move downward from the vertical UAV [93] while it was tilted to the side of the fuselage, resulting in off-target depositions.

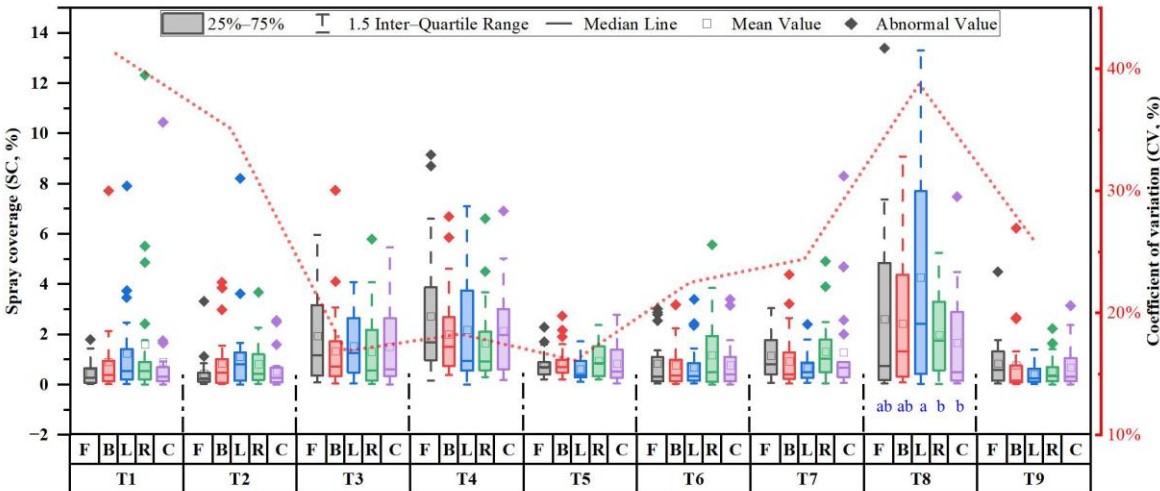

**Figure 11.** Deposition characteristics of SC in the horizontal direction of the canopy. F represents the front of the pear tree canopy, B represents the back, L represents the left, R represents the right, and C represents the middle of the canopy. Different lowercase letters indicate significant differences at $p \leq 0.05$. The box plot represents DD, and the dashed line represents CV.

**Table 9.** Characteristics of CV_P in the horizontal direction of the canopy.

| | DD | | | | SC | | | |
|---|---|---|---|---|---|---|---|---|
| | **Factor SV** | **Factor FS** | **Factor FH** | **Factor FD** | **Factor SV** | **Factor FS** | **Factor FH** | **Factor FD** |
| Optimum Level | SV3 | FS3 | FH2 | FD3 | SV2 | FS3 | FH3 | FD3 |
| Order | | FD > SV > FS > FH | | | | FH > SV > FS > FD | | |

### 3.2.4. Droplet Deposition Characteristics in the Vertical of the Canopy

As mentioned earlier, the **FT orchard** model of pear orchards has two main layers in the canopy, one being NL, which is located in the upper part of the canopy, and the other is FL, which is located in the lower part of the canopy. The NL and FL are subdivided into two sides, respectively. And there are four layers from top to bottom of the canopy, which we label L1, L2, L3, and L4.

As shown in Figure 12, the main differences in DD were between the NL and FL layers in all tests. The NL layer was significantly higher than the FL layer ($p < 0.05$), but there were no differences between L1 and L2 of the NL layer nor between L3 and L4 in the FL layer.

Only three groups of L4 layers (T4, T7, T8) met the requirement of 25 deposits/cm$^2$, the deposition of these groups were 28.3 deposits/cm$^2$, 26.77 deposits/cm$^2$, and 68.65 deposits/cm$^2$. While the L4 layer of all the other groups did not meet the minimum control requirements.

The CV_V of DD met the criteria of T2, T4, T5, T7, T8, of which the CV_V of T4, T7, T8 were 34%, 35%, and 21%, respectively. The PC of DD fluctuated in a small range with values between 8% and 21%. The PC of three tests (T4, T7, T8) was 15%, 17%, and 21%, respectively.

The deposition trend of SC was like the pattern of DD (Figure 13). In contrast, the SC of the lowest layer (L4) was only higher than 1% in T4 and T8.

There were wide fluctuations in the CV_V of SC at different canopy heights, which ranged from 11% to 55%, for groups T2, T5, T7, and T8 at 29%, 11%, 30%, and 30%, all of which were below 40%. It should be mentioned that the CV_V of T4 was 46%, which did not meet the requirement.

The PC of SC fluctuated between 8% and 21%. T2, T5, T7, and T8 had a PC of 15%, 21%, 15%, and 17% in that order, where T5 also had the highest PC of all the tested groups.

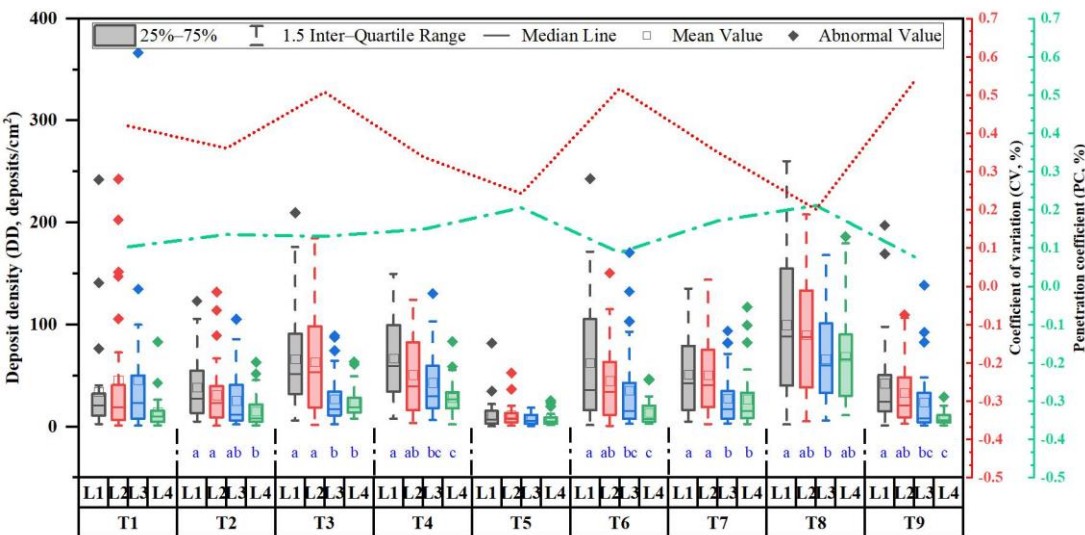

**Figure 12.** Deposition characteristics of DD in the vertical direction of the canopy. L1 and L2 represent the upper and lower layers of the nutrient layer (NL), and L3 and L4 represent the upper and lower layers of the fruit layer (FL). Different lowercase letters indicate significant differences at $p \leq 0.05$. The box plot represents DD, and the dashed line represents CV, the dotted line represents PC.

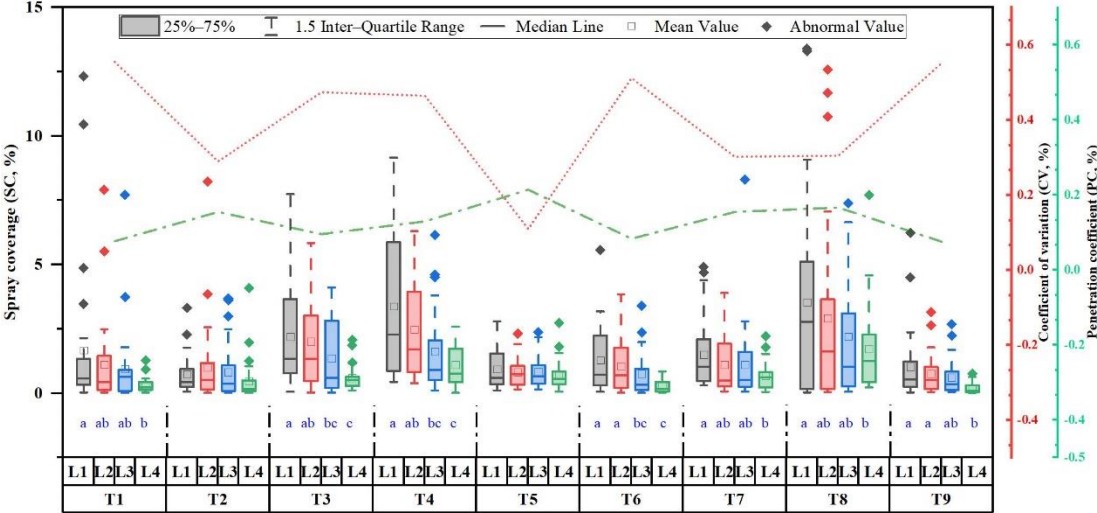

**Figure 13.** Deposition characteristics of SC in the vertical direction of the canopy. L1 and L2 represent the upper and lower layers of the nutrient layer (NL), and L3 and L4 represent the upper and lower layers of the fruit layer (FL). Different lowercase letters indicate significant differences at $p \leq 0.05$. The box plot represents DD, and the dashed line represents CV, the dotted line represents PC.

The extreme difference analysis showed that the factors that affected the CV_V of DD and SC were FS > SV > FD > FH, FS > FH > SV > FD in descending order. The best combination was SV2 (SV3)-FS2-FH3-FD3, SV2-FS2-FH3-FD2, respectively. The effects of different factors on the PC of DD and SC were FS > FH > FD > SV, FS > FH > SV > FD in this order. The best combination was SV3-FS2-FH1-FD3, and SV2-FS2-FH3-FD3, respectively. As shown in Table 10.

Regardless of whether DD or SC was used as an evaluation index, factor FS was the key factor affecting the dispersion of droplet deposition and uniformity of penetration in the vertical direction of the canopy [94]. It was similar to previous studies [53] that investigated the optimal droplet distribution control parameters for citrus trees using UAV and Taguchi methods. Among the control parameters discussed, flight speed had the most significant effect, with a contribution of 74.0%.

**Table 10.** Characteristics of CV_V and PC in the vertical direction of the canopy.

| | | DD | | | | SC | | | |
|---|---|---|---|---|---|---|---|---|---|
| | | **Factor SV** | **Factor FS** | **Factor FH** | **Factor FD** | **Factor SV** | **Factor FS** | **Factor FH** | **Factor FD** |
| CV_V | Optimum Level | SV2 (SV3) | FS2 | FH3 | FD3 | SV2 | FS2 | FH3 | FD2 |
| | Order | FS > SV > FD > FH | | | | FS > FH > SV > FD | | | |
| PC | Optimum Level | SV3 | FS2 | FH1 | FD3 | SV2 | FS2 | FH3 | FD3 |
| | Order | FS > FH > FD > SV | | | | FS > FH > SV > FD | | | |

Because the NL and FL layers have different structures and the NL layer was sparser in the upper part of the canopy, the deposition decreased sequentially downward. At the same time, more droplets were concentrated in the upper layer of the canopy [95]. There was a better uniformity of droplet deposition in the horizontal direction than in the longitudinal direction in this study, which was partly due to the weaker longitudinal penetration of the hexacopter UAV [96]. Another main reason was the barrier effect of the larger canopy [97].

### 3.3. Response Surface Analysis

As observed in Figures 14 and 15, the general tendency was the same whether evaluated with DD or SC, i.e., In order to achieve greater droplet deposition, the operational parameters in this study needed to be optimally set as follows: along the tree rows, the lower the flight height, the slower the flight speed, and the higher the application rate.

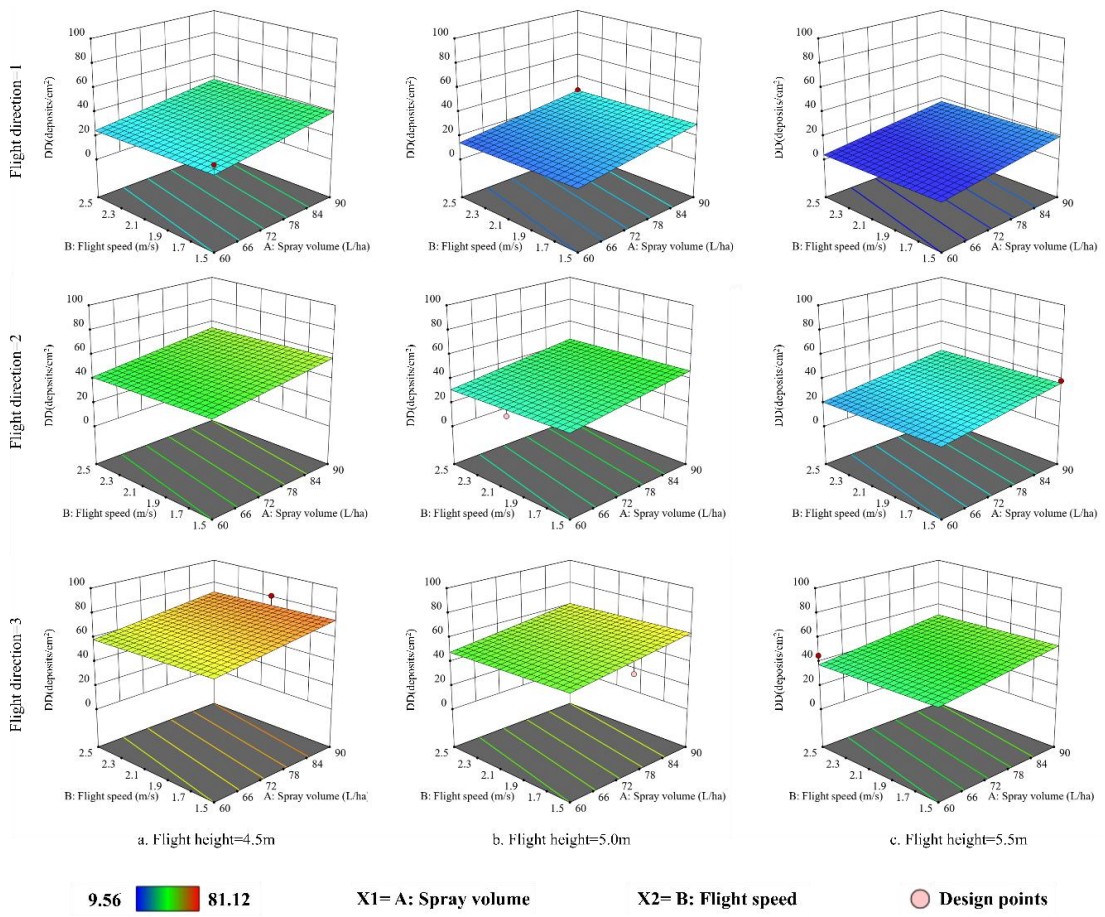

**Figure 14.** Response surface analysis of DD.

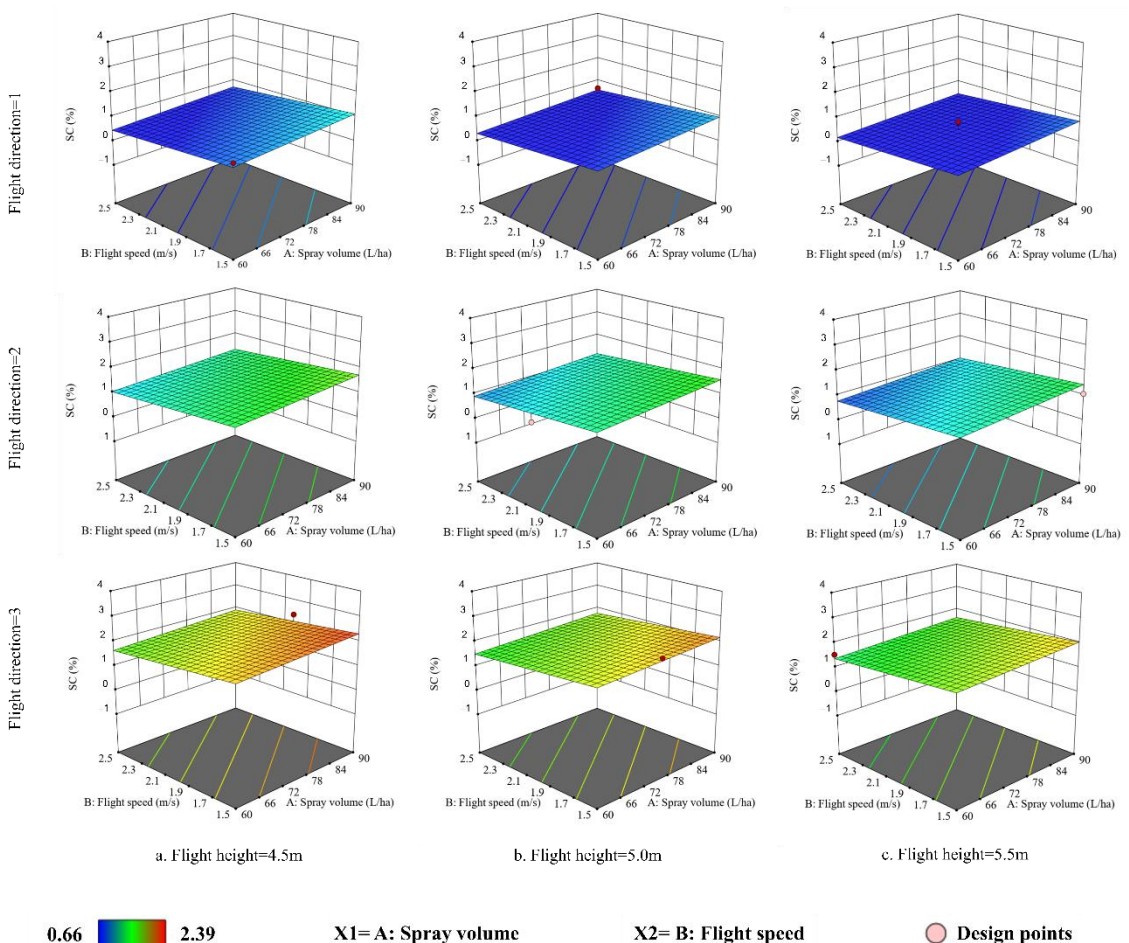

**Figure 15.** Response surface analysis of SC.

The results of the response surface analysis were consistent with the results of the extreme difference analysis, showing that the best combination to improve SC was SV3-FS1-FH1-FD3, and the best combination to improve DD was SV3-FS1-FH1-FD3. The effect surface analysis also showed that SC and DD were negatively correlated with FS and FH and positively correlated with SV. It was also not difficult to find out that the observation of this regularity had been widely confirmed. Although there was no problem with the conclusions obtained from the extreme difference analysis and response surface analysis, it was not necessarily the optimal and economical choice for pesticide application. It was simply the option with the highest droplet deposition within the set parameters.

### 3.4. Comprehensive Evaluation of Spray Quality

The indicators, DD, SC, and CV, were evaluated jointly by Equations (3)–(6), and the results Com_E are shown in Figure 16. In order of Com_E values, the top three were T8, T4, and T3.

Synthesizing the previous analysis of DD and SC at different locations, except that the SC on the abaxial of the leaves cannot meet the requirements, the groups that can meet the DD and SC protection standards at different positions on the adaxial of the leaves in the horizontal and vertical directions were T8 and T4.

The analysis of different corresponding surfaces showed that the slower the flight speed and the lower the flight height, the higher the application rate, which was conducive to promoting the deposition of droplets. Therefore, the operation in the pear orchard when the parameters of the UAV cannot be lower than the value of T4 and T8. Namely, it was required to fly along the tree rows with an application rate of no less than 75 L/ha, a speed of no more than 2 m/s, and a height of no more than 5 m.

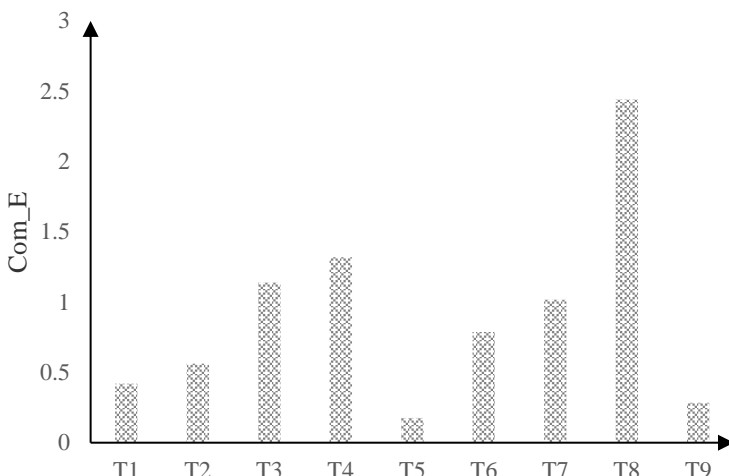

**Figure 16.** Comprehensive evaluation of the quality of droplet deposition. T1 to T9 represent different test groups.

The fact is that pesticides sprayed by UAVs require extra care as they can easily drift to damage non-targeted organisms and can contaminate the environment [98,99]. Therefore, when using UAVs, it was important to not only ensure that they satisfied pest efficacy requirements but there was also a key recommendation to determine the minimum application rate that would provide reliable pest efficacy while maintaining operational efficiency [100].

In this study, we used WSP to visualize the differences in UAV spraying for different operational parameters. We included both DD and SC in the spray quality assessment. However, it was an evaluation result based on an empirical basis and criteria. The practical application of pesticides in the field was a complex process. Therefore, it was recommended for the follow-up study to include pest efficacy in the evaluation at the same time. This will provide a comprehensive understanding of the field performance of the UAV when spraying fruit trees with a dense canopy. This holistic approach links pesticide residue, deposition, coverage, and canopy penetration data to pest efficacy, which provides more information than the pseudo-quantitative data from image-based WPSs interpreted in isolation.

## 4. Conclusions

By means of ANOVA and principal component analysis, factors FS (flight speed) and FD (flight direction) had the greatest effect on droplet coverage ($p < 0.05$), and the different factors were influenced by FD > FS > FH > SV in that order. Factor SV (spray application volume rate), FH (flight height), and FD had a significant effect on deposit density, and the relationship between the effects was Factor FD > FH > SV > FS in order.

The extreme difference analysis and the response surface analysis showed that the highest deposit density and spray coverage could be obtained by the combination SV3-FS1-FH1-FD3, which means flight along the row with an application rate of 90 L/ha, the flight speed of 1.5 m/s, flight height of 4.5 m.

The parameters suitable for operation at T4 (application rate of 75 L/ha, flight speed of 1.5 m/s, flight height of 5 m, flight along the row) and T8 (application rate of 90 L/ha, flight speed of 2 m/s, flight height of 4.5 m, flight along the row) were derived from the analysis of all test groups. Namely, it was required to fly along the tree rows with an application rate of no less than 75 L/ha, a speed of no more than 2 m/s, and a height of no more than 5 m. This satisfied the control requirements in different horizontal and vertical directions.

Although different parameter settings can change the deposition, there was still no significant improvement on the abaxial side of the leaves, which can only meet the demand of deposit density control but not spray coverage control. The coverage of the abaxial side of the leaves was less than 1%, with a maximum of only 0.84%, which appeared in the T4.

For UAV spraying, or this top-to-bottom spraying, the deposition on the abaxial side of the leaves is more important. It was found that by satisfying the control needs on the abaxial side of the leaves, the needs on the adaxial side of the leaves could be satisfied.

In this study, a 1% evaluation index was used as the criterion to meet the spray coverage, and it was found that a deposit density of 25 deposits/cm$^2$ could be met as long as the spray coverage criterion was met. The spray coverage better reflects the evaluation effect of spraying, but whether to use 1% as the evaluation standard needs a lot of experimental verification, which provides a reference for future standard settings.

In order to measure the effect of droplet deposition more accurately, it was subsequently necessary to combine the pest efficacy, find the spray coverage index suitable for low volumes, and use multiple indices for a comprehensive comparative analysis. At the same time, it was not possible to simply use the extreme difference analysis method to measure orchards with different canopy structures. This should be analyzed depending on the characteristics of the canopy structure and the location where spraying was most difficult.

**Author Contributions:** Conceptualization, P.Q. and L.Z.; methodology, P.Q., L.Z. and H.H.; software, P.Q., L.Z. and T.L.; validation, P.Q., L.Z., and C.W.; formal analysis, P.Q., L.Z., M.H. and Z.H.; writing—original draft preparation, P.Q. and L.Z.; writing—review and editing, P.Q., L.Z and Z.W.; supervision, Y.L., X.H., and J.M. All authors have read and agreed to the published version of the manuscript.

**Funding:** This work was carried out due to financial support from the Major Program of Hubei Agricultural Science and Technology Innovation Center (2020-620-000-002-05), the World Bank "Climate Smart Staple Crop Production Project" (CWB Pro No. 144531/GEF Pro No. 5221), the National Key Research and Development Program of China (2020YFD1000202), China Agriculture Research System (No. CARS-28-20), Deutsche Forschungsgemeinschaft (DFG, German Research Foundation)-328017493/GRK 2366 (Sino-German International Research Training Group AMAIZE-P), Sanya Institute of China Agricultural University Guiding Fund Project, Grant No. SYND-2021-06, and the 2115 talent development program of China Agricultural University.

**Data Availability Statement:** Not applicable.

**Acknowledgments:** The authors acknowledge language editing by Sabine Nugent. Special thanks to Sabine Nugent for suggesting corrections to the article. The work was conducted in close collaboration with the Sino-German research project Pattern Management in China (PMC)—a holistic approach for sustainable, site-specific agriculture in China.

**Conflicts of Interest:** The authors declare no conflict of interest. The funders had no role in the design of the study; in the collection, analyses, or interpretation of data; in the writing of the manuscript; or in the decision to publish the results.

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
