# Peer review of "Effect of Operational Parameters of Unmanned Aerial Vehicle (UAV) on Droplet Deposition in Trellised Pear Orchard"

_drones, doi:10.3390/drones7010057_

Round 1

Reviewer 1 Report

General comments

This paper proposes an experimental design in order to optimise deposition on pear orchards with an UAV. Since the interest in this application technique is rising globally, this paper represents an interesting contribution.

 The experimental strategy is based on a Tagushi Orthogonal Array principle. It allows to limit the number of tests (reduce from 81 to 9) considering that each variable is theoretically independent from others. This important hypothesis is not entirely satisfied in this case since the SV volume is dependent on FS (and also FH)… The more independent variable would have been the spray liquid flowrate… or, if not monitored, the pressure or at least the number of nozzles activated.

 Most of the studied parameters are not normalized by the theoretical application volume. As a result, the more it is sprayed, the more is recovered (SC and DD)… so 2 of the 4 variables are bind to this statement. This questions the adequacy of the experimental design.

 Most of the parameters are mostly qualitative, SC is an indirect estimation of the applied dose recovery but is not demonstrated with a calibration.

 It is not clear the Materials and Methods that samplers will be oriented horizontally or vertically and what samplers will be used in the first part of the results. It is then less clear where do the general results come from all samples or only a group (vertical or horizontal) ?? Please clarify. It makes the results quite redundant…

Detailed comments

Line 48-49: % relates to what exactly ? What is total output?  This sentence is not clear.  

 Line 89-90 : This sentence, as introductive, shall be placed earlier (line 73?)

 Line 185: What is the droplet size range for these nozzles ?

 Line 204 : What means a valid replicate exactly ?

 Line 205 Table 2. It is not obvious the spray swath width was considered according to the flight height ? This table shall be more discussed since It is understood from Table 3 that all factors are not totally independent? This shall be demonstrated considering the number of nozzles activated and related pressure.

The last factor is not randomly operated and cannot be considered as independent.

 Line 253 : was this Spray Coverage correlated with a dosage since this parameter (as well as DD and Dv50) are mostly qualitative indication?

 Line 295 : SC33% means 1/3 of the rated dose in this case ??

 Line 319-320 : this equation 6 is hardly understandable without explanation. CV and SC are related to the same variable (CV is just an indication of the variability of SC). This equation seems to be based on the total independence of DD, CV and SC that can be discussed…

Line 345 : the presentation of results in Table 7 is a bit surprising since all variables settings were grouped into modalities (T1 to T9)… It looks obvious that SV3 allows the highest deposit since it corresponds to the highest application rate…

The predominance of FD is to be explained since this parameter combines different values of other parameters… Why don’t we direct switch to P or V in this case?... Could it be possible to have quantitative values for interactions, to better explain this table 7 ?

 Line 359 : it shall be mentioned the threshold of significance

 Line 362: What is Factor D exactly here ?  can FD be considered as other variables (since it is a mix of them? …) Maybe this last row of table 8 shall be removed ?

 Line 383 : It would have been useful for the reader to provide raw data.

As an example the average SC values are:  FD3 = 2.08, FD2 = 0.89 and FD1 = 0.83

Average DD values are : FD 3 = 59.7; FD2 = 35.3; FD1 = 24.0

Average DV50 values are: FD3 = 146; FD2 = 176 and FD1 = 153…

 Line 386 : formally, WSF provide impact sizes.

 Line 408 : isn’t it the case for UASS applications in général ?

 Line 430 : Is this horizontal (and further vertical) sampling explained before?

 It would probably be needed to aggregate all the results in a more clear way ?

 Line 534 : The benefit of such figures is not obvious.

Author Response

Dear Ms. Brigitta Katler and dear reviewers:

Re: Manuscript ID: drones-2112321 and Title: Effect of operational parameters of unmanned aerial vehicle (UAV) on droplet deposition in trellised pear orchard.

Thank you for your letter and the reviewers’ comments concerning our manuscript entitled “Effect of operational parameters of unmanned aerial vehicle (UAV) on droplet deposition in trellised pear orchard" (drones-2112321). Those comments are valuable and very helpful. We have read through the comments carefully and have made corrections. Based on the instructions provided in your letter, we uploaded the file of the revised manuscript. Revisions in the text are shown using the red highlight for additions. The responses to the reviewer's comments are marked in red.

We would love to thank you for allowing us to resubmit a revised copy of the manuscript and we highly appreciate your time and consideration.

Best Regards.

Reviewer 2 Report

Review paper for ” Effect of operational parameters of unmanned aerial vehicle (UAV) on droplet deposition in trellised pear orchard”

To Authors;

This paper discusses the effect of the drone flying parameters on the droplet deposition in the pear orchard.  This study brings an opportunity for a novel approach for the orchard environment with a systematic configuration of the trees layout and arrangement. Drones spraying is an important tool and popular technology in addressing the labor issues in the agriculture sector. The findings show an important aspect from drone spraying operation point of view using WSP, however the efficacy test may be needed to verify the effective effect on the pest or disease of interest in this orchard.  This manuscript can be improved by mentioning the time of test, since the temporal effect due to climatic conditions and stability are the key parameters in pest and disease spraying operation. Authors also need to carefully examine the term used in describing drone spraying performances.  The specific comments and may required justification as follows;

Line 285 - “However, pesticide UAV sprayers typically apply 7.5-450 L/ha to fruit trees, which is classified as a low application rate for tree crops according to the standards [69,70]”. The spraying rate for drones rarely goes beyond 50L/ha per application or operation. Does this number imply UAV and ground application? 

Line 34- 25/cm2 upper scripts for cm2

Line 292 - please change UAS to UAV in order to have consistency in the manuscript.

Line 319 - in each of the equations, what does the ‘i’ represent? 

Line 320 - Eq. 6 needs to remove the space between CV and V and CV and P. This should be the same as written on the upper section in Line 270. 

Line 321-325 - please delete this section, which is not part of the the manuscript

Line 462 - what is the meaning of the wind field? Are the authors meant for downwash or air turbulence under the drone rotors? 

Line 468 - “resulting in different depositions on the left 468 and right” can be rephrase to “off-target depositions’

Line 531-533- Both Figures need to be enlarged, please make sure the font is readable. 

Line 553 - “application volume of no less than 75L’ why not using the application rate in L/ha? 

Line 568 - “include pest control effects”do you mean the pest efficacy test? 

Line 582 - “spray volume of 90L/ha”This is not spray volume, the term is application rate. Similarly, you may need to check across your entire manuscript using the right term.

Author Response

Dear Ms. Brigitta Katler and dear reviewer:

Re: Manuscript ID: drones-2112321 and Title: Effect of operational parameters of unmanned aerial vehicle (UAV) on droplet deposition in trellised pear orchard.

Thank you for your letter and the reviewers’ comments concerning our manuscript entitled “Effect of operational parameters of unmanned aerial vehicle (UAV) on droplet deposition in trellised pear orchard" (drones-2112321). Those comments are valuable and very helpful. We have read through the comments carefully and have made corrections. Based on the instructions provided in your letter, we uploaded the file of the revised manuscript. Revisions in the text are shown using the red highlight for additions. The responses to the reviewer's comments are marked in red.

We would love to thank you for allowing us to resubmit a revised copy of the manuscript and we highly appreciate your time and consideration.

Best Regards.

Reviewer 3 Report

This is an interesting study adding some answers to the critical questions of what is the appropriate spray volume for UAV applications in trees, what is the appropriate speed and height and how should the UAV move into the field in order to achieve the best performance. The authors performed a comprehensive experiment examining all the above parameters and present some detailed results of the affects. The experimental design and the analysis approach are scientifically sound and complete. I enhance the authors to continue their studies by examining the actual effects, besides the findings depicted in WSP that are presented in the present research.

I propose only some minor corrections in the text, as discussed below. Thereafter, I suggest the acceptance and publication of the present article.

Lines 72-73  UAVs carrying multispectral or hyperspectral cameras for detection (of what?)

Line 83 “lower penetration coefficient, which also tends to produce pesticide drift” Is this true? How is this established? Is there any literature?

Lines 85-86. “many locations limit the use of ground spraying machinery, such as hilly 85 terrain, high-density planting patterns, irregular spacing and fragmented land” But the authors refer to a completely mechanized system based on a trellised scheme. So their research doesn’t focus on such kind of areas.

Line 114: “1.7 m·s-1” Please use a uniform form for units along the text. In Line 32 for instance it is referred as m/s. I would suggest an SI metric system for all the variables.

Line 184. “..and relative plant height”. I think the authors mean “relative flight height”

Line 272 “to the total number of droplets in this direction” Please explain how the total number of droplets was estimated.

Formula 3. I think its incorrect. It might be (DD1-25)(50-25)

Lines 321-335 are authors guidance from MDPI and should be deleted.

Line 553. Please correct to “….no less than 75l/ha”

Line 575 [lease correct “factors FB” to “factors FS”

Author Response

(The authors gave the same response as above.)

Round 2

Reviewer 1 Report

Authors have considered all comments and provided appropraite corrections. This paper is now acceptable for publication